

# Dental characters used in phylogenetic analyses of mammals show higher rates of evolution, but not reduced independence

Neil Brocklehurst[1] and Gemma Louise Benevento[2]

[1] Department of Earth Sciences, University of Oxford, Oxford, United Kingdom
[2] School of Geography, Earth and Environmental Sciences, University of Birmingham, Birmingham, United Kingdom

## ABSTRACT

Accurate reconstructions of phylogeny are essential for studying the evolution of a clade, and morphological characters are necessarily used for the reconstruction of the relationships of fossil organisms. However, variation in their evolutionary modes (for example rate variation and character non-independence) not accounted for in analyses may be leading to unreliable phylogenies. A recent study suggested that phylogenetic analyses of mammals may be suffering from a dominance of dental characters, which were shown to have lower phylogenetic signal than osteological characters and produced phylogenies less congruent with molecularly-derived benchmarks. Here we build on this previous work by testing five additional morphological partitions for phylogenetic signal and examining what aspects of dental and other character evolution may be affecting this, by fitting models of discrete character evolution to phylogenies inferred and time calibrated using molecular data. Results indicate that the phylogenetic signal of discrete characters correlate most strongly with rates of evolution, with increased rates driving increased homoplasy. In a dataset covering all Mammalia, dental characters have higher rates of evolution than other partitions. They do not, however, fit a model of independent character evolution any worse than other regions. Primates and marsupials show different patterns to other mammal clades, with dental characters evolving at slower rates and being more heavily integrated (less independent). While the dominance of dental characters in analyses of mammals could be leading to inaccurate phylogenies, the issue is not unique to dental characters and the results are not consistent across datasets. Molecular benchmarks (being entirely independent of the character data) provide a framework for examining each dataset individually to assess the evolution of the characters used.

## INTRODUCTION

Accurate reconstructions of phylogenetic relationships are essential for studying the evolutionary history of a clade, with hypotheses being based on molecular or morphological data, or both. While it is comparatively straightforward to observe patterns of evolution in molecular sequence data and therefore develop models more closely representing the

Corresponding author
Neil Brocklehurst,
neil.brocklehurst@earth.ox.ac.uk

evolutionary processes, this is more difficult in the case of morphological characteristics due to a poorer understanding of how novel morphology is evolved from ancestral traits. Nonetheless, morphological data are our only means of reconstructing the phylogenetic relationships of fossil organisms that are too old to preserve DNA or usable proteins. It is therefore imperative that we strive to better understand the evolutionary modes of morphological traits. In recent years many studies have examined how variation in evolutionary patterns of discrete morphological traits, not accounted for by current analyses, may be affecting phylogenetic inferences (e.g., *O'Keefe & Wagner, 2001*; *Scotland, Olmstead & Bennett, 2003*; *DeGusta, 2004*; *Sansom, Wills & Williams, 2017*; *Billet & Bardin, 2018*).

The high percentage of dental characters used in the reconstruction of fossil mammal phylogenies has become a particular cause for concern. Numerous studies have highlighted issues such as the non-independent evolution of dental characters (*Kangas et al., 2004*; *Kavanagh, Evans & Jernvall, 2007*; *Harjunmaa et al., 2014*; *Dávalos et al., 2014*; *Billet & Bardin, 2018*) and increased convergence relative to other character partitions due to ecological selective pressures (*Evans et al., 2007*; *Kavanagh, Evans & Jernvall, 2007*). In a recent meta-analysis, *Sansom, Wills & Williams (2017)* examined the phylogenetic signal of tooth and osteological character partitions, using phylogenies derived from molecular data as a benchmark. This study found that osteological characters were more consistent with the molecularly-derived phylogenies and contained greater phylogenetic signal than dental characters. Further, parsimony analyses with only dental characters produced results less similar to the molecularly-derived phylogenies than analyses where the same number of characters were selected at random from both partitions (*Sansom, Wills & Williams, 2017*).

This paper builds on the work of *Sansom, Wills & Williams (2017)* in two principal ways. *Sansom, Wills & Williams (2017)* employed two partitions, dental and osteological, to assess the performance of dental characters relative to osteological characters in phylogenetic analyses. As such, while dental characters have been demonstrated to be problematic, an understanding of whether this problem was limited to them, or whether it extends to other partitions, is lacking. We therefore examine phylogenetic signal in six morphological partitions in mammals in order to establish whether any other skeletal regions may be a poor indicator of phylogeny.

Secondly, we also aim to understand why dental characters may be producing phylogenies less congruent with molecularly-derived benchmarks. Many studies have established that morphological characters frequently violate at least some of the principal assumptions of parsimony (see below): between-character rate homogeneity (all characters being just as likely to transition), within-character rate homogeneity (all character states within the same character being similarly likely to transition), and character independence. We test each morphological character partition for variation in these parameters.

In most published phylogenetic analyses performed using parsimony, the characters are weighted equally (*Kälersjö, Albert & Farris, 1999*; *Kluge, 2005*; *Goloboff et al., 2008*). Under such a scheme, a change in any character is given equal influence in determining tree length. However, such a scheme only produces robustly supported results when the characters are all equally likely to change. If, however, there is variation in the rates of

character evolution, certain characters will change more frequently and are more likely to show homoplasy (*Felsenstein, 1981*; *Goloboff, 1993*). While parsimony analysis does not incorporate an explicit evolutionary model, an equal weights analysis does rely on equal between-character rates for its accuracy.

Furthermore, in most published phylogenetic analyses, transitions between different combinations of character states are given equal weight (i.e., a transition from state 0 to state 1 is just as likely as a transition from state 1 to state 0; an assumption of within-character rate homogeneity). This assumption may be relaxed by incorporating step matrices which give greater weight to particular transitions (*Sankoff & Cedergren, 1983*), or by ordering (*Fitch, 1971*), an extreme modification of step matrices, setting the possibility of non-adjacent transitions to 0. However, step matrices are rarely employed, and the use of ordered characters is still heavily debated (see *Marjanović & Laurin, 2019* for summary of their history), so most analyses assume equality of within-character rates.

Finally, all methods of phylogenetic analysis (parsimony, Bayesian, and likelihood), treat all characters as independent of one another (i.e., an assumption that a change in one character will have no effect on the transition probability in another character). Extensive study has shown this assumption of independence to be frequently violated (e.g., *Kangas et al., 2004*; *Kavanagh, Evans & Jernvall, 2007*; *Harjunmaa et al., 2014*; *Dávalos et al., 2014*; *Billet & Bardin, 2018*), with many traits or regions forming integrated modules that change as a unit (*Goswami, 2006*; *Goswami, 2007*; *Goswami & Polly, 2010*).

By analysing phylogenetic signal, between- and within-character rates, and character independence across six morphological partitions, within mammals as a whole and within four mammalian subclades, we aim to better understand how morphological characters can be selected and formulated during phylogenetic analyses of mammals. The results should provide future studies that intend to reconstruct the relationships of fossil mammals with a framework to enable more evidence-based decisions about which characters are more reliable for use in phylogenetic analyses.

## MATERIALS AND METHODS

### Data

This study builds on the protocol established by *Sansom, Wills & Williams (2017)*, where molecularly-derived phylogenies are used as the framework over which morphological evolution may be analysed. This allows the evolutionary patterns of the characters to be examined over a phylogeny produced and time calibrated from data entirely independent of those characters. Unlike *Sansom, Wills & Williams (2017)*, all phylogenies used were also scaled to time using molecular data, as required by the phylogenetic comparative methods employed. For mammals the time-scaled molecularly-derived phylogeny was taken from *Dos Res et al. (2012)*, and the morphological data from *Bi et al. (2014)*, both recent and comprehensive datasets. Although the *Bi et al. (2014)* matrix was focussed on Mesozoic mammals, it contains a broad sampling of modern clades, including taxa from the monotremes, marsupials and placentals. The morphological characters were divided between six partitions: dental, cranial, axial, forelimb (including pectoral girdle), hindlimb

(including pelvic gridle), and soft tissue. Taxa not present in both the morphological matrix and molecularly-derived tree were dropped. If, after doing so, a character showed no variation in score among the remaining taxa, that character was also dropped from subsequent analyses.

As well as the global analysis of mammals, four subclades were subjected to the same analyses to test for variation in the macroevolutionary patterns within Mammalia. The clades chosen were as follows: Artiodactyla (Molecularly-derived tree from *Hassanin et al. (2012)*, Morphological matrix from *Spaulding, O'Leary & Gatesy (2009)*), Carnivora (Molecularly-derived tree from *Eizirik et al. (2010)*, Morphological matrix from *Tomiya (2010)*), Primates (Molecularly-derived tree from *Perelman et al. (2011)*, Morphological matrix from *Ni et al. (2013)*) and Marsupialia (Molecularly-derived tree from *Mitchell et al. (2014)*, Morphological matrix from *Beck (2017)*). These clades were chosen for the following reasons: (1) they have been analysed using morphological character matrices containing characters from all six of the morphological partitions; (2) there exist time calibrated molecularly-derived phylogenies with substantial taxonomic overlap with the morphological matrices; (3) the character list, data matrix and time calibrated phylogeny were available in usable formats; and (4) they are morphologically and ecologically diverse lineages, and therefore the morphological characters have the potential to be heavily influenced by functional and ecological constraints.

## Phylogenetic signal

Levels of homoplasy relative to the molecularly-derived phylogeny were used as an estimate of the phylogenetic signal of the characters, measured using Pagel's lambda (*Pagel, 1999*), a metric shown to perform well under simulations (*Münkemüller et al., 2012*). This statistic produces a value between 0 and 1, where 0 indicates that character states are distributed independent of phylogeny (no phylogenetic signal). Other methods of calculating phylogenetic signal in discrete characters, for example Moran's I (*Gittleman & Jot, 1990*) or Fritz & Purvis's D (*Fritz & Purvis, 2010*), were not used as they are only suitable for binary characters and would require a large proportion of the characters to be dropped. For each character, taxa scored as unknown were dropped from the tree. If more than a quarter of the taxa were scored as unknown, the character was not considered in this or subsequent analyses. Pagel's lambda was calculated in R version 3.3.2 (*R Core Team, 2016*) using the *fitDiscrete* function in the package geiger (*Harmon et al., 2007*).

## Testing the assumptions of phylogenetic analysis

Within-character rate homogeneity was tested by fitting models of discrete character evolution to the observed phylogeny and trait values using the function *fitDiscrete* in the R package geiger. This method calculates the likelihood of a particular model based on the data, and also estimates the values of variable parameters within the model that best fits the observed data (*Pennell & Harmon, 2013*; *Pennell et al., 2014*). Two models were compared: an equal rates (ER) model, where every possible character state transformation has the same rate, and an all-rates-different (ARD) model, where every possible character state transformation is allowed a different rate. The models are compared using the Akaike

information criterion, which penalises the parameter-rich ARD model. The Akaike weights of the ER model are used as a metric to assess how well a character obeys the assumption of within-character rate homogeneity.

The *fitDiscrete* function also allows testing of between-character rate homogeneity. As mentioned above, as well as identifying the model of discrete character evolution that best fits the trait and phylogeny, it also identifies the rates of character-state transformation that best fits the observed data. A higher rate of change means a character is more likely to change multiple times by convergence. If a character was found to best fit the ER model in the above analysis, then the single rate of change was assigned to the character. If the ARD model was found to fit best, the rate assigned to that character was the mean of all rates assigned to each possible transformation, weighted by the number of times each transformation occurred over the phylogeny. The number of transitions was inferred by stochastically mapping the character over the phylogeny 1000 times using the *make.simmap* function in the R package phytools (*Revell, 2012*), and calculating the mean frequency of each possible transition.

To test character independence, the method of *Pagel (1994)* was applied to pairwise comparisons of characters. This is again a model-fitting approach, where non-independent and independent models of character evolution are fit to pairs of traits and the observed phylogeny. Under the non-independent model, the rate of character change in trait 1 will depend on which character state is observed in trait 2, and vice versa. Under the independent model, both characters change state independently of each other. Again, the two models may be compared via the Akaike information criterion, and the Akaike weights of the independent model may be used as a metric for how well a pair of characters obeys the assumption of independent evolution. Unfortunately, this method is only applicable to binary characters, so non-binary characters were not considered in this section of the analyses. The analysis was implemented using the function *fitPagel* in phytools.

## Statistical comparisons

Pagel's lambda values for each character partition were compared using generalised least squares (GLS), using the R package nlme (*Pinheiro et al., 2017*). For each partition, a null model where all the phylogenetic signal of all partitions comes from the same distribution, was compared to a model where only the partition of interest had a different phylogenetic signal (H1). The Akaike weights was used to infer which best fit the data. Partitions that better fit the H1 model were deemed to have significantly different phylogenetic signals than the other partitions, with the GLS coefficient used to identify whether higher (positive coefficient) or lower (negative coefficient). The same method was also applied to the rate values, the support for the ER model, and support for the independent model of evolution.

The rate of character change for each character, and the Akaike weight for the ER model for each character, were both compared to Pagel's lambda using the Kendall's tau correlation coefficient, a non-parametric method that does not assume normality of the data. This latter test could not be applied to the Akaike weights values of the independent model of evolution because these represent pairwise comparisons of characters rather than individual characters. The number of characters in partitions was compared to the median

phylogenetic signal, median rate, median support for the ER model and median support for the independent model to test the influence of character sampling on the results.

## RESULTS

### Results from the total Mammalia dataset

The median phylogenetic signal calculated from the *Bi et al. (2014)* character matrix (the total Mammalia dataset) was 1 for all partitions (white points, Fig. 1A). This indicates that at least half of the characters in each partition are synapomorphies for a single clade. The dental characters do show a larger range and interquartile range of lambda values than the other partitions. However, the range of values observed for cranial characters is similar to that of dental characters. In the GLS analysis, cranial characters are the only partition to have significantly lower phylogenetic signal than other partitions (Table 1).

Dental characters show no evidence of increased within-character rate heterogeneity than do the other partitions (Fig. 1B). In fact, the Akaike weights of the equal rates (ER) model for dental characters are the highest of all the partitions, and in the GLS analysis no partitions have significantly better support for the ER model than other partitions (Table 2). Dental characters also show no evidence of increased non-independence relative to other partitions (Fig. 1C). Only the forelimb partition was found to have significantly worse support for the independent model of evolution than other partitions (Table 3). The hindlimb was found to have significantly better support for the independent model.

However, dental characters have the highest median rates of evolution compared to all other partitions (Fig. 1D), and the increase in rates is significant according to the GLS analysis (Table 4). No other partitions were found to have a significant difference in rate relative to the null model.

### Results from mammalian subclade datasets

The artiodactyl datasets produced similar results to those of mammals overall, albeit with considerably more variation in phylogenetic signal from the vertebral, forelimb and soft tissue characters (Fig. 2A). The dental characters are the only partition where the GLS analysis found phylogenetic signal to be significantly reduced relative to other partitions (Table S1). Rates of dental evolution are again significantly higher than for other partitions (Fig. 2D, Table S4). There is no significant difference found between the Akaike weights support for the ER model of evolution in teeth (Table S2), nor the support for the independent model of character evolution, compared to other partitions (Table S3). The skull partition shows better support for the independent model, while the forelimb shows statistically significantly reduced independence.

The carnivoran dataset also found dental characters to have significantly lower phylogenetic signal than other partitions (Fig. 3A, Table S7). In this clade the dental character partition also has higher rates than all other partitions except the forelimb (for which there is only one character) (Fig. 3D).

The primate and marsupial datasets produced results conflicting with the other two subclades and mammals as a whole (Figs. 4 and 5). The dental partition in primates has significantly higher phylogenetic signal and significantly lower rates of evolution than other

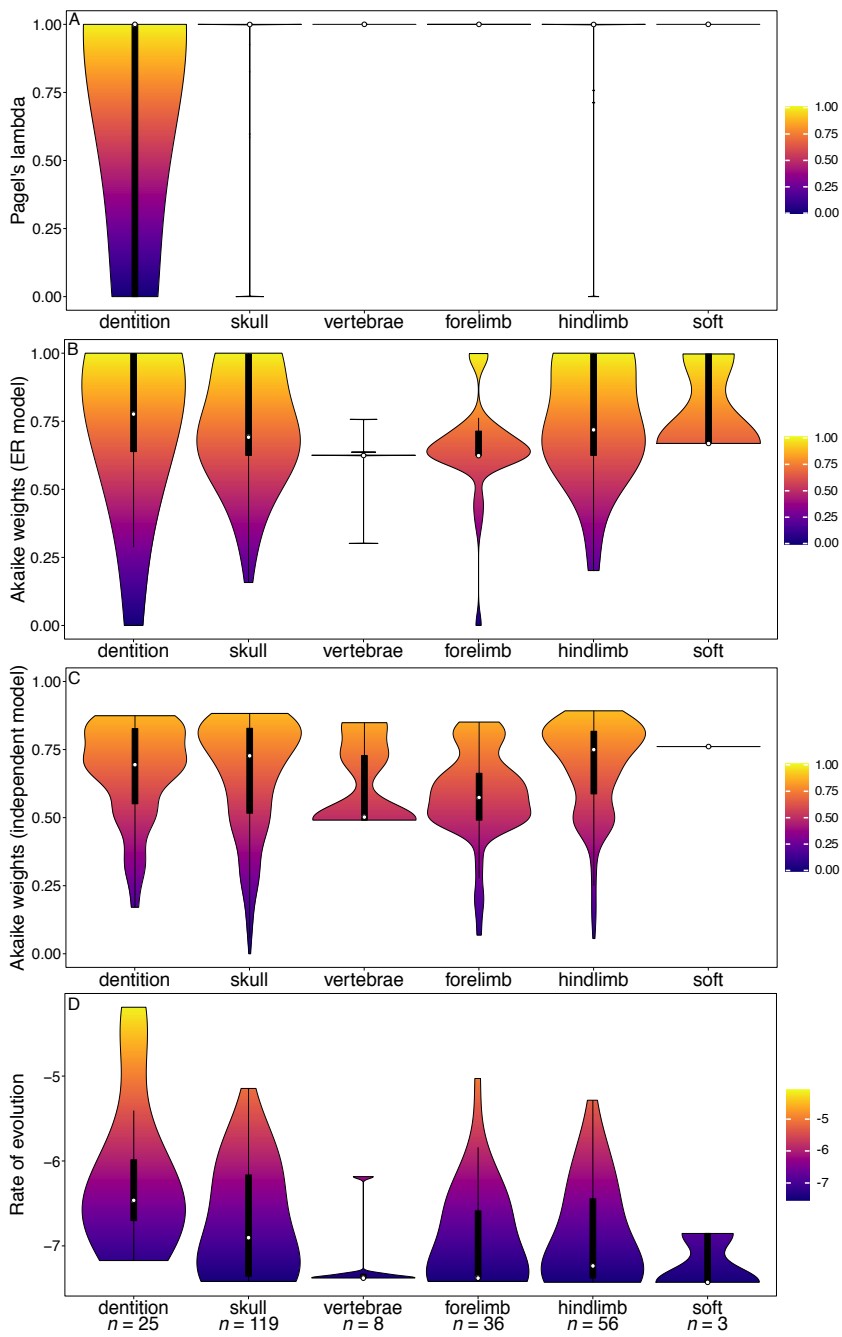

**Figure 1** **Violin plots illustrating results from the *Bi et al. (2014)* character matrix (total Mammalia).**
(A) Pagel's lambda values (phylogenetic signal) of each character. A value of 0 indicates no phylogenetic signal, while a value of 1 indicates high phylogenetic signal. (B) Akaike weights support for the ER model of evolution of each character. Characters with an Akaike weights score of 1 have equal rates of within-character evolution between each state, while characters with a score of 0 display unequal rates of within-character state evolution. (C) Akaike weights support for the independent model of evolution of all pair-wise comparisons of characters in each partition. Pairwise comparisons that have an Akaike weights score

**Figure 1 (...continued)**
of 1 evolve independently of one another, while pairwise comparisons with a score of 0 display character non-independence. (D) Rates of character evolution of each character (log10 transformed). The number of characters in each partition can be found at the base of the figure ($n = X$). For each partition, the horizontal spread of the violin plot represents the density of data at each point on the $y$-axis. Box plots with a white point representing the median are plotted within each violin plot. The heatmap is a visual representation of the $y$-axis.

**Table 1  Results of GLS analyses of Pagel's λ (phylogenetic signal of character partitions) in mammals.** Rows coloured are those where the partition best fits the H1 model (partition has a different lambda value to all others); blue indicates lower phylogenetic signal, red indicates higher phylogenetic signal.

| Partition | Median λ | GLS Coefficient | lnL (null) | lnL (H1) | AIC (null) | AIC (H1) | Akaike weights (null) | Akaike weights (H1) |
|---|---|---|---|---|---|---|---|---|
| Teeth | 1 | −0.15 | −103.02 | −102.6 | 210.0 | 211.1 | 0.63 | 0.37 |
| Skull | 1 | −0.11 | −103.02 | −101.4 | 210.0 | 208.8 | 0.35 | 0.65 |
| Vertebrae | 1 | 0.15 | −103.02 | −103.5 | 210.0 | 212.9 | 0.81 | 0.19 |
| Forelimb | 1 | 0.16 | −103.02 | −101.3 | 210.0 | 208.5 | 0.32 | 0.68 |
| Hindlimb | 1 | 0.07 | −103.02 | −103.8 | 210.0 | 213.6 | 0.86 | 0.14 |
| Soft tissue | 1 | 0.15 | −103.02 | −103.4 | 210.0 | 2129 | 0.81 | 0.19 |

**Table 2  Results of GLS analyses of Akaike weight support for the equal rates (ER) model of character evolution in mammals.**

| Partition | Median weight | GLS Coefficient | lnL (null) | lnL (H1) | AIC (null) | AIC (H1) | Akaike weights (null) | Akaike weights (H1) |
|---|---|---|---|---|---|---|---|---|
| Teeth | 0.78 | 0.02 | 26.77 | 24.75 | −49.55 | −43.50 | 0.95 | 0.05 |
| Skull | 0.71 | 0.02 | 26.77 | 24.45 | −49.55 | −42.89 | 0.97 | 0.03 |
| Vertebrae | 0.62 | −0.13 | 26.77 | 26.60 | −49.55 | −47.19 | 0.76 | 0.24 |
| Forelimb | 0.62 | −0.09 | 26.77 | 26.91 | −49.55 | −47.82 | 0.70 | 0.30 |
| Hindlimb | 0.72 | 0.04 | 26.77 | 24.85 | −49.55 | −43.70 | 0.95 | 0.05 |
| Soft tissue | 0.67 | 0.05 | 26.77 | 25.69 | −49.55 | −45.38 | 0.89 | 0.11 |

partitions (Fig. 4, Tables S9, S12). The dental partition also has significantly better support for the equal rates model of evolution than other partitions. However, primate characters suffer from being highly integrated: all partitions other than the vertebrae and soft tissue characters show a significantly low fit to the independent model of evolution (Fig. 4C, Table S11).

In marsupials, while many of the character partitions, including dentition, show a wide range of Pagel's lambda values, the lambda values of the tooth characters are more concentrated towards higher values compared other partitions (Fig. 5A). The tooth characters show no significant difference in their phylogenetic signal relative to other partitions (Table S13). The dental characters show no significant difference from any other partitions in support for the ER model of evolution (Table S14), and no significant difference in rates (Fig. 5D). In contrast to the other datasets, however, the marsupial dataset does support increased character non-independence of dental characters relative to other partitions, with median Akaike weights support for the independent model of evolution lower than all other partitions (Fig. 5C; Table S15).

**Table 3  Results of GLS analyses of Akaike weight support for the independent model of character evolution in mammals.** Rows coloured are those where the partition best fits the H1 model (partition has a different rate value to all others); blue indicates lower Akaike weights, red indicates higher.

| Partition | Median weight | GLS Coefficient | lnL (null) | lnL (H1) | AIC (null) | AIC (H1) | Akaike weights (H0) | Akaike weights (H1) |
|---|---|---|---|---|---|---|---|---|
| Teeth | 0.69 | 0.012 | 1,043.0 | 1,040.5 | −2,082 | −2,075 | 0.97 | 0.10 |
| Skull | 0.73 | 0.013 | 1,043.0 | 1,040.8 | −2,082 | −2,076 | 0.76 | 0.22 |
| Vertebrae | 0.50 | −0.034 | 1,043.0 | 1,041.1 | −2,082 | −2,076 | 0.95 | 0.04 |
| Forelimb | 0.57 | −0.077 | 1,043.0 | 1,073.1 | −2,082 | −2,140 | ∼0 | ∼1 |
| Hindlimb | 0.75 | 0.039 | 1,043.0 | 1,050.7 | −2,082 | −2,095 | 0.001 | 0.999 |
| Soft tissue | 0.76 | 0.100 | 1,043.0 | 1,042.4 | −2,082 | −2,079 | 0.83 | 0.16 |

**Table 4  Results of GLS analyses of rates of character evolution in mammals.** Rows coloured are those where the partition best fits the H1 model (partition has a different rate value to all others); red indicates higher rate.

| Partition | Median rate | GLS Coefficient | lnL (null) | lnL (H1) | AIC (null) | AIC (H1) | Akaike weights (null) | Akaike weights (H1) |
|---|---|---|---|---|---|---|---|---|
| Teeth | 0.0016 | 0.29 | −53.01 | −43.63 | 110.03 | 93.27 | 0.0002 | 0.99 |
| Skull | 0.0010 | 0.02 | −53.01 | −55.19 | 110.03 | 116.38 | 0.96 | 0.04 |
| Vertebrae | 0.0006 | −0.20 | −53.01 | −52.59 | 110.03 | 111.19 | 0.64 | 0.36 |
| Forelimb | 0.0006 | −0.08 | −53.01 | −53.76 | 110.03 | 113.52 | 0.85 | 0.15 |
| Hindlimb | 0.0007 | −0.07 | −53.01 | −53.85 | 110.03 | 113.70 | 0.96 | 0.04 |
| Soft tissue | 0.0006 | −0.21 | −53.01 | −53.12 | 110.03 | 112.24 | 0.75 | 0.25 |

## Correlation tests

In all five datasets, there is a negative correlation between lambda and rate of character evolution, significant in all except Carnivora (Table 5). The correlation between the lambda values and Akaike weights of the ER model is weaker in all five, but in some is still significant. None of the parameters tested correlate significantly with number of characters in each partition (Table 6).

## DISCUSSION

Mammalian tooth characters have been a source of much discussion over the last two decades, due in part to their dominance of the character lists used in morphological phylogenetic analyses of mammals, itself to an extent a product of their dominance in the mammalian fossil record. Teeth have been shown to suffer from issues such as large amounts of homoplasy (*Evans et al., 2007*; *Dávalos et al., 2014*) and non-independence (*Kangas et al., 2004*; *Harjunmaa et al., 2014*). While these issues clearly do impact on the utility of dental characters in phylogenetic analysis, what has received less attention is whether dental characters are in fact worse affected than other body partitions in these regards. The majority of studies cited above focus solely on teeth, but issues of homoplasy due to ecological and functional constraints might be expected to affect other character partitions (e.g., limb characters being functionally linked to locomotion). Indeed, ecological constraint and developmental linkage has been demonstrated in cranial and limb

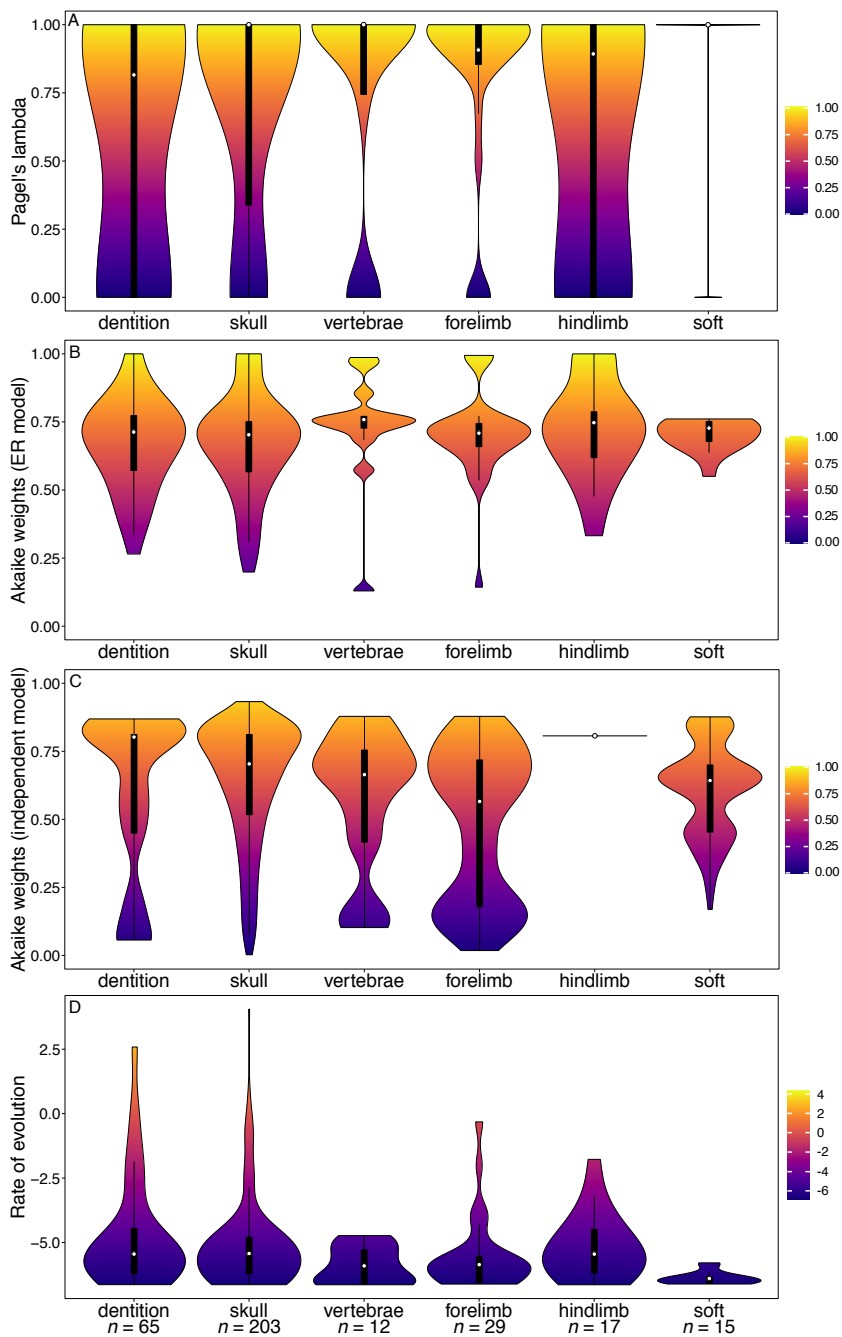

**Figure 2** **Violin plots illustrating results from the *Spaulding, O'Leary & Gatesy (2009)* matrix (Artiodactyla).** (A) Pagel's lambda values (phylogenetic signal) of each character. (B) Akaike weights support for the ER model of evolution of each character. (C) Akaike weights support for the independent model of evolution of all pairwise comparisons of characters in each partition. (D) Rates of character evolution of each character (log10 transformed). The number of characters in each partition can be found at the base of the figure ($n = X$).

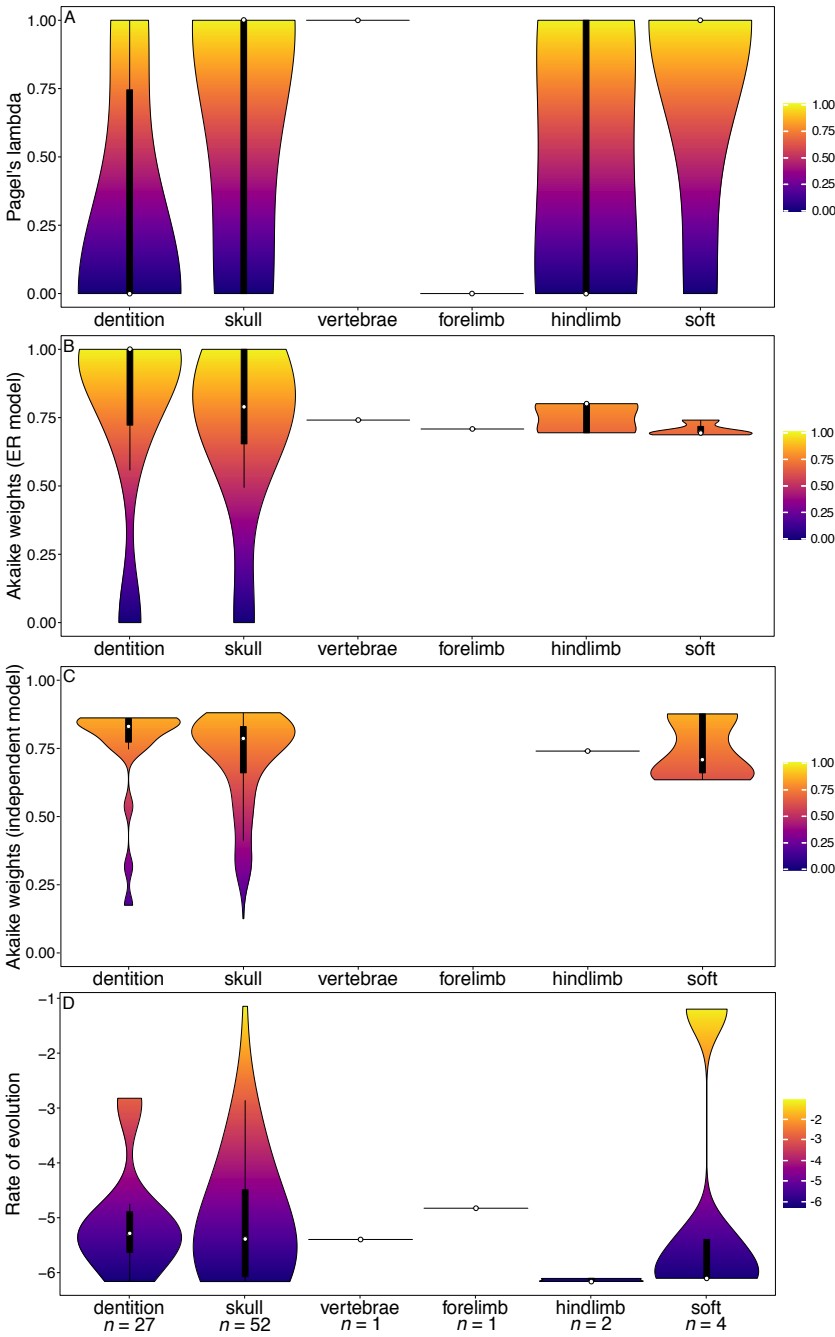

**Figure 3** **Results from the *Tomiya (2010)* matrix (Carnivora).** (A) Pagel's lambda values (phylogenetic signal) of each character. (B) Akaike weights support for the ER model of evolution of each character. (C) Akaike weights support for the independent model of evolution of all pairwise comparisons of characters in each partition. (D) Rates of character evolution of each character (log10 transformed). The number of characters in each partition can be found at the base of the figure ($n = X$).

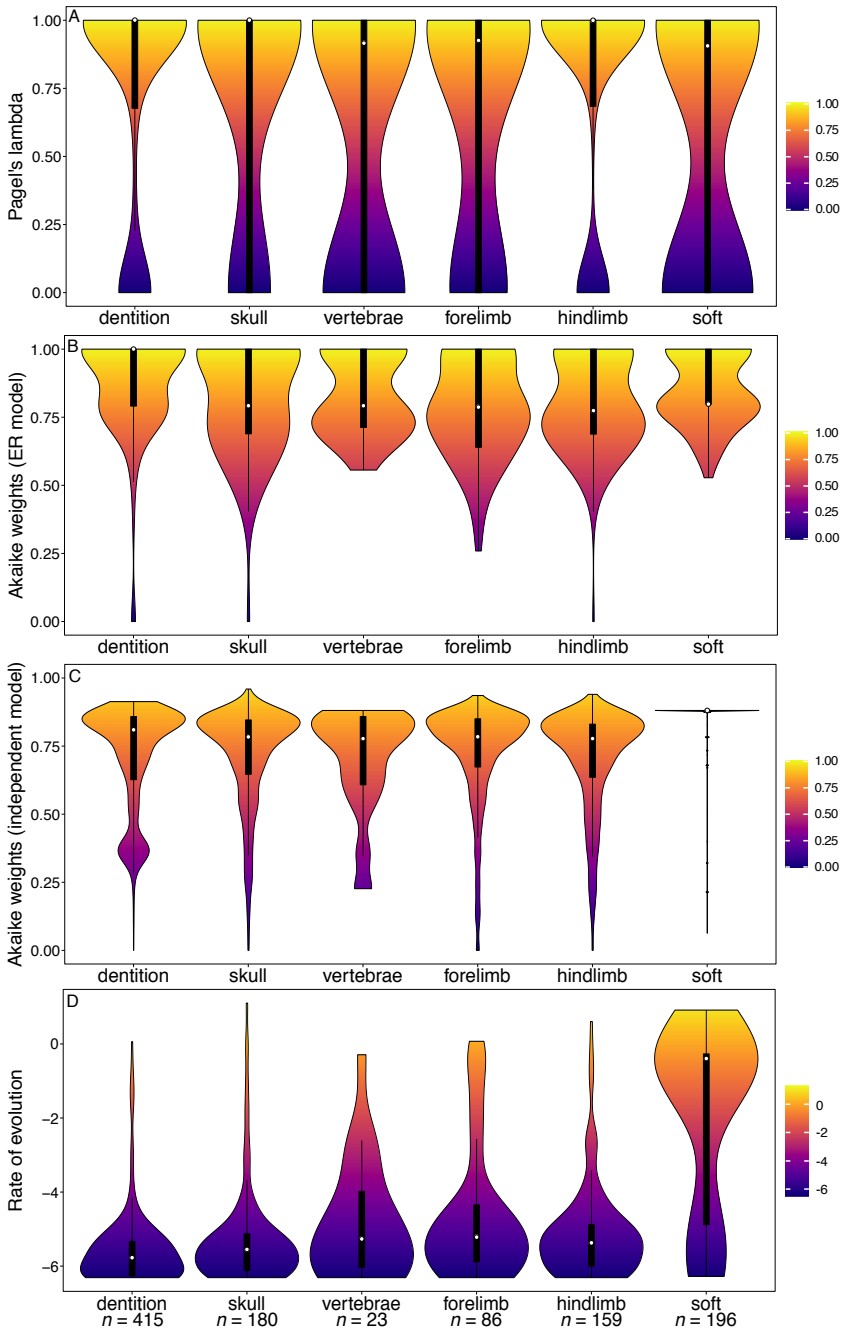

**Figure 4  Violin plots illustrating results from the *Ni et al. (2013)* matrix (Primates).** (A) Pagel's lambda values (phylogenetic signal) of each character. (B) Akaike weights support for the ER model of evolution of each character. (C) Akaike weights support for the independent model of evolution of all pairwisee comparisons of characters in each partition. (D) Rates of character evolution of each character (log10 transformed). The number of characters in each partition can be found at the base of the figure (*n* = *X*).

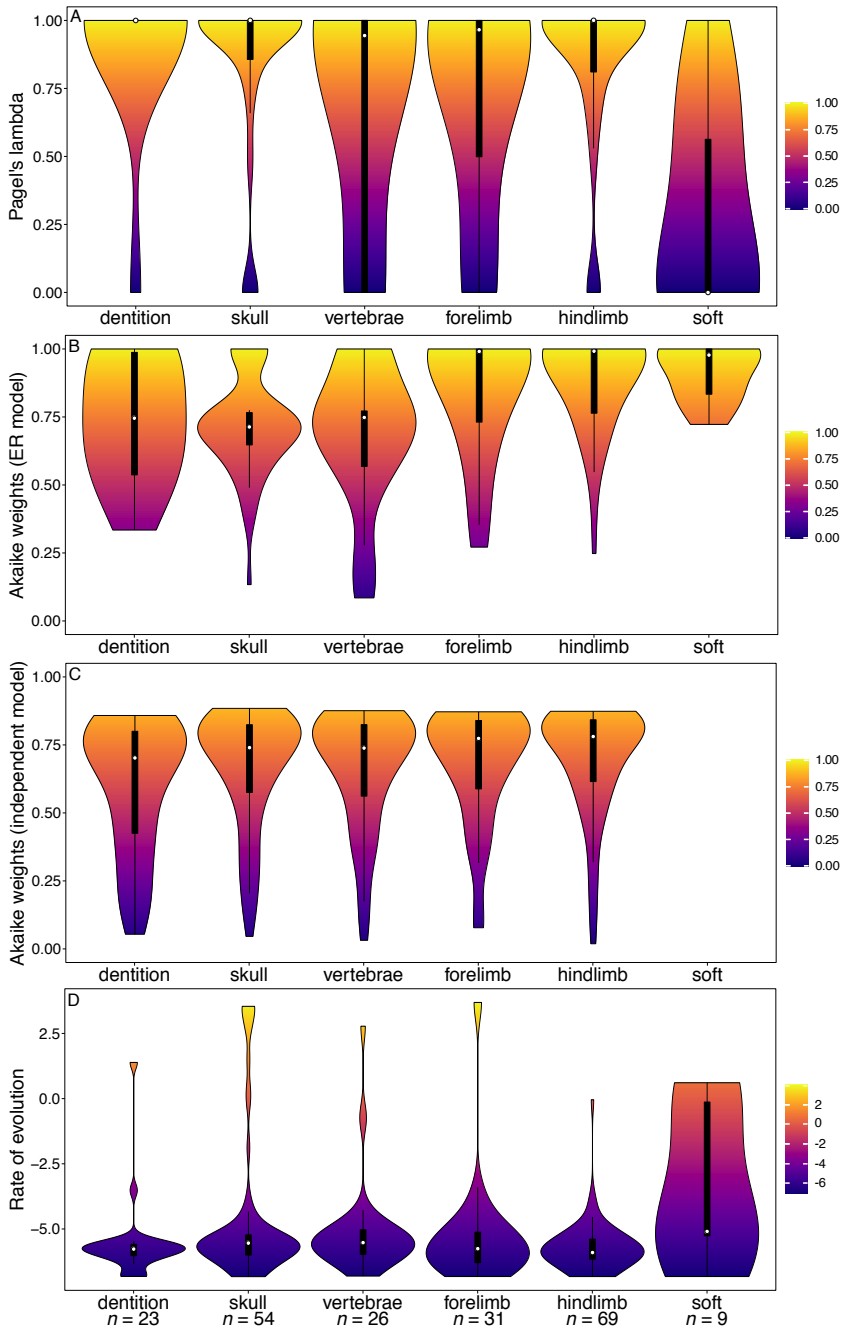

**Figure 5** **Violin plots illustrating results from the *Beck (2017)* matrix (Marsupialia).** (A) Pagel's lambda values (phylogenetic signal) of each character. (B) Akaike weights support for the ER model of evolution of each character. (C) Akaike weights support for the independent model of evolution of all pairwise comparisons of characters in each partition. (D) Rates of character evolution of each character (log10 transformed). The number of characters in each partition can be found at the base of the figure (*n* = *X*).

**Table 5 Results of Kendal's tau correlation tests between rates of evolution and support for the equal rates model, and phylogenetic signal.**

| | Pagel's lambda vs Rates of character evolution | Pagel's lambda vs Akaike weight support for ER model of character evolution |
|---|---|---|
| Total mammal dataset | $-0.22$ ($p = 3.67 \times 10^{-6}$) | $-0.050$ ($p = 0.2996$) |
| Artiodactyl dataset | $-0.24$ ($p = 3.49 \times 10^{-10}$) | $0.15$ ($p = 1.05 \times 10^{-4}$) |
| Carnivoran dataset | $-0.1$ ($p = 0.4435$) | $-0.04$ ($p = 0.5701$) |
| Primate dataset | $-0.22$ ($p \leq 2.2 \times 10^{-16}$) | $-0.012$ ($p = 0.56$) |
| Marsupial dataset | $-0.22$ ($p = 2 \times 10^{-5}$) | $0.11$ ($p = 0.025$) |

**Table 6 Results of Kendal's tau correlation tests between number of characters in the partitions and phylogenetic signal, support for the equal rates and independent models, and rates of evolution.**

| Correlation test | Kendall's tau | P value |
|---|---|---|
| Number of characters in dataset partition $\sim$ Median Pagel's lambda | $-0.009$ | 0.95 |
| Number of characters in dataset partition $\sim$ Median Akaike weights (ER model) | 0.31 | 0.10 |
| Number of characters in dataset partition $\sim$ Median Akaike weights (independent model) | 0.26 | 0.17 |
| Number of characters in dataset partition $\sim$ Median rate | 0.28 | 0.12 |

characters across various tetrapod groups, including mammals (*Ruvinsky & Gibson-Brown, 2000*; *Young & Hallgrímson, 2005*; *Sadleir & Makovicky, 2008*). The same argument could be made for the issue of character non-independence: while this has been demonstrated to be a problem with mammal dentition, recent work on modularity and integration highlights that this issue might just as strongly impact on non-dental characters (*Goswami, 2006*; *Goswami, 2007*; *Goswami & Polly, 2010*).

Our analyses suggest that increased homoplasy driven by increased rates of evolution may affect dental characters to a greater extent than other partitions. Dental characters from the total Mammalia dataset and the artiodactyl and carnivoran datasets are found to evolve at faster rates than the other character partitions, and so are more likely to transition multiple times. The strong and significant inverse correlations between phylogenetic signal and rates of evolution in all tested datasets indicates that rate variation is likely to be the main driving force behind loss of phylogenetic signal, more so than within-character rate heterogeneity. However, this signal is not consistent across all the tested clades. In the marsupial and primate datasets, dental characters have lower rates (and higher phylogenetic signal) than most other partitions.

Moreover, while the results obtained here seem to suggest that dental characters have lower phylogenetic signal than some other characters when optimised over a molecular-based phylogeny, they are not alone in this respect. The total Mammalia dataset indicates that cranial characters also have low phylogenetic signal. In both primates and marsupials, the soft tissue characters have lower Pagel's lambda values than any other partition (Figs. 4A, 5A) and in carnivorans both limb partitions perform poorly in this respect (Fig. 3A).

The results observed in artiodactyls raise a possibility that might warrant future study: the increase in rates of dental evolution observed might be due to the dominance of herbivores in this dataset. Herbivory has been suggested to be a driver of dental disparity in mammals (*Jernvall, Hunter & Fortelius, 1996*; *Jernvall, Hunter & Fortelius, 2000*) as their morphology tracks a constantly changing resource (plants). Since the functional requirements of eating meat has not changed over time, carnivorous mammals show reduced dental disparity and less evolutionary change (*Van Valkenburgh, 1988*; *Wesley-Hunt, 2005*). In an analysis of diversification patterns across all mammals, herbivores showed significantly higher diversification rates than carnivores or omnivores (*Price et al., 2012*). While this analysis focussed on lineage diversification, the authors cited increased specialisation and niche-subdivision as a potential driving force behind diversification patterns, and morphological diversification patterns should respond to these drivers in the same way.

It is finally worth noting that in the total-Mammalia dataset and two of the three placental subclades tested, there is little evidence that tooth characters are affected by non-independence to any greater extent than the other morphological partitions. The primate and marsupial datasets are the exception, with dental characters showing a weaker fit to the independent model than all other partitions. The integration of the dental characters and their low rates of evolution in marsupials may be due to their unusual development: neonatal marsupials, born extremely early in their development need to attach to the teat, leading to precocial development of the jaw and facial region in marsupials (*Smith, 1996*; *Smith, 2006*). This could lead to this region evolving as a more integrated module. Alternatively, it may be a result of character selection: the *Beck (2017)* dataset contains large numbers of characters relating to the presence or absence of particular dental loci in both upper and lower jaws, likely to be heavily integrated.

## CONCLUSIONS

The concept pioneered by *Sansom, Wills & Williams (2017)*, of testing morphological discrete characters over a molecular benchmark, is a powerful tool, and it would be highly recommended that researchers studying clades where molecularly-derived phylogenies exist examine the performance of their characters in this manner. But given the extremely wide variation in results found by this study, where different partitions produced different relative phylogenetic signals (with the primates and marsupials in particular producing results conflicting strongly with the other datasets studied), one should perhaps be cautious of basing assumptions of character quality on the results of large meta-analyses. While the latter are useful for identifying broad-scale patterns, it is necessary that each dataset be examined individually, and decisions made based on the macroevolutionary patterns observed in that clade. However, larger-scale meta-analyses do have the advantage in that they are less likely to be affected by idiosyncrasies of individual datasets and the choices/intentions of the researchers. For example, the total-Mammalia dataset includes only three soft-tissue characters relating to the different modes of reproduction, likely intended to separate monotremes, marsupials and placentals, and therefore this is not representative of the total variation in soft tissue evolution. A detailed examination of

specific characters and how variation in their formulation affects the results would be an avenue for future research.

A fair and comprehensive sampling of characters across partitions should be the aim; experiments incorporating random sampling of characters show that sampling across partitions leads to a more reliable estimation of phylogenetic relationships than sampling within single partitions (*Pattinson et al., 2015*). While dental characters have been shown to suffer from issues of homology and non-independence (*Kangas et al., 2004*; *Evans et al., 2007*; *Harjunmaa et al., 2014*), the comparison of the dental characters to finer partitions of data presented here demonstrates that these issues are not unique to teeth. In fact, in some cases other regions perform even worse, and the nature of these issues varies from clade to clade.

## ACKNOWLEDGEMENTS

We would like to thank Roger Benson and Robert Sansom for helpful discussion. Robert Asher, Robin Beck and an anonymous reviewer provided helpful comments on an early draft of the manuscript.

### Funding

Neil Brocklehurst's research is funded by Deutsche Forschungsgemeinschaft grant number BR 5724/1-1. Gemma Louise Benevento's research is funded by a NERC studentship from the Oxford DTP in Environmental Research (NE/L0021612/1), alongside additional support provided by the European Research Council (grant agreement 637483). The funders had no role in study design, data collection and analysis, decision to publish, or preparation of the manuscript.

### Grant Disclosures

The following grant information was disclosed by the authors:
Deutsche Forschungsgemeinschaft: BR 5724/1-1.
NERC studentship from the Oxford DTP in Environmental Research: NE/L0021612/1.
European Research Council: 637483.

### Competing Interests

The authors declare there are no competing interests.

### Author Contributions

- Neil Brocklehurst conceived and designed the experiments, performed the experiments, analyzed the data, prepared figures and/or tables, authored or reviewed drafts of the paper, and approved the final draft.
- Gemma Louise Benevento conceived and designed the experiments, prepared figures and/or tables, authored or reviewed drafts of the paper, and approved the final draft.

## Data Availability

The characters, phylogenetic trees and details of character use in the analyses are available in Data S1–S15. Analysis code is available in Data S16. Supplementary results are presented in the Tables S1–S16.

## Supplemental Information

Supplemental information for this article can be found online at http://dx.doi.org/10.7717/peerj.8744#supplemental-information.

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
