# Peer review of "Dental characters used in phylogenetic analyses of mammals show higher rates of evolution, but not reduced independence"

_PeerJ, doi:10.7717/peerj.8744_

## Round 0.1 · original submission · Major Revisions

Dear authors,

Given the number of comments from the three reviewers, I have accepted the decision of ‘major revisions’.

Whether you wish to change the datasets you are working on (as suggested) or to expand upon that number is up to you. However, if you can provide the matrices and R scripts as suggested (either as supplementary material or in on of the online repositories).

I look forward to receiving your revised manuscript.

·

Basic reporting

see general comments

Experimental design

see general comments

Validity of the findings

see general comments

Additional comments

review of Brocklehurt & Benevento MS
"Dental characters used in phylogenetic analyses of mammals show higher rates of evolution, but not reduced independence"
PeerJ
Robert J. Asher, October 2019

This is an interesting & important paper which I'd like to see published. I've made a number of comments below which I hope help the authors to improve the delivery.

I'd be delighted if the authors were to better articulate the standard by which they measure accuracy. As expressed in my comments below pertaining to lines 69-72, 81 & 124, the authors perpetuate the awful habit in the literature of confusing "molecular" with "true". What the authors really mean is "well corroborated" and they should be clear about this (see below).

Intuitively, authors who select few dental characters may be choosing those that are more obvious, less variable, & easier to code vs. authors who include many dental characters, but this study doesn't really quantify how much of the overall character sample in a given study is represented by each partition (teeth, forelimb, cranium, etc.). The intuition for mammalian teeth is that they're diagnostic at species level, easily preserved, and are only present in the mouth so are typically atomized into relatively more characters per unit volume of phenotype compared to other partitions. This study is better than most in beginning to disentangle the relevant factors, for example by by narrowing the subject of comparison and not comparing teeth to (for example) "the skeleton". One could go farther still to try and get some parity; e.g., if an ankle or astragalus were atomized like teeth (many characters in small phenotypic volume), perhaps they too exhibit the various, negative properties often claimed in the literature, although the authors understandably do not want to recode morphology from these past studies and are limited by the extent to which past authors coded characters in one region or another. Nonetheless, could the authors state for each study (Beck et al., Pattinson et al., Bi et al., etc.) what proportions of the whole dataset each partition contributes? E.g., Beck et al. 2014 (basis for Beck's 2017 study) has about 25 dental out of 258 total characters, whereas Pattinson et al. 2015 (derived originally from Seiffert 2009) has 184 dental & jaw characters out of a total of 360. Does having only 10% vs. 50% dental contribute to any of the (even small) differences in signal in terms of rate, homoplasy, etc? Relatedly I'd be interested to know if the authors believe that the studies they've cited differ in character rate, signal, etc. due to something intrinsic to each partition & taxa, or because a given study (say Bi et al. vs. Beck et al.) used different criteria to select their characters.

The essence of their results are represented by the four figures, but for most readers these will be very difficult to interpret. Each has a Y axis value, a heat-map and horizontal thickness, but it's not at all clear to me what the heat map or thickness of each graphic are supposed to represent. It would be great if the authors could explain in plain English why (for example) all partitions except dentition & skull for the Bi et al. matrix appear to have little variation in Pagel's lambda (all w/ horizontal bars at 1.0, I guess median based on line 207 but the ingredients of each figure should be clear from its caption). In addition to a fuller explanation of what these figures mean in the main text, please try to make the captions more self-contained & accessible without forcing your readers to flip back-n-forth from Methods & Results; make clear what the units are and what the various axes are (including horizontal thickness).

Also the authors do not discuss at all a key discovery dating to the 1990s, namely that for any problem of a given partition in isolation, combination across partitions ameliorates bias and can reveal hidden support (Gatesy & Baker 2005 Syst Bio; Gatesy et al. 1999 Cladistics; Thompson et al. 2012 J Zoo Sys Evol Res). A key finding of Pattinson et al. 2015 was that incomplete fossils known for characters across morphological partitions were slightly but significantly better reconstructed than similarly complete fossils known for just one or few partitions. Surely this is relevant & worth discussion in this paper, which concludes (line 343) little more than the obvious platitude that one should "be cautious of basing assumption [sic] of character quality on the results of large meta-analyses".

abstract:
line 28: "...relationships. [new sentence] HOWEVER, variation in their evolutionary modes"

line 56: "morphological data ARE..."

lines 69-72: Other papers use "morphological" vs "molecular" as misleading adjectives for what's really meant, something like "potentially biased" vs "true". Here the authors have repeated the phrase "molecular tree", as if this were synonymous with "true tree" (although in fairness they're just paraphrasing other authors). Rather than repeating this sloppiness, just use adjectives like "molecular" to describe data, not trees, as the the authors have on occasion correctly done elsewhere (e.g., lines 50-52), but not lines 120-124, 132, 134, 138, 145, 338-39.

line 81: "molecular benchmark" is better than "molecular tree", but the authors could do even better by identifying a topology that is "well corroborated", as close to "true" as the philosophers will let us write. More to the point, some molecular datasets yield demonstrably wrong topologies (e.g., "the guinea pig is not a rodent"), whereas others show extraordinary congruence with completely different bodies of phylogenetic data, from embryology to the fossil record (e.g., monophyly of Sarcopterygia, Placentalia, Primates, etc.). In order to pass judgement on a given partition, the "benchmark" the authors need is actually a well-corroborated tree, these days usually but not always based on molecular data. The term "benchmark" does not do justice to this concept so write "well-corroborated tree" (which by the way is similar to but not identical with most of the topologies figured by Meredith et al. 2011 main text AND suppdata).

line 82: Is rate homogeneity really an "assumption" of parsimony, or do you really mean a feature of data that could yield statistical inconsistency when using parsimony as an optimality criterion? Analogously MP doesn't assume that all branches are short but can still be misled by accumulations of nonhomologous similarities on long branches. Note on the flip side that probabilistic assumptions of homogenous rates for taxa & characters proportional to branch length are also susceptible to error (see Goloboff et al. 2018 Palaeontology or Smith et al. 2018 Biol Letters).

lines 91-95: I don't think this is true. Your phrase "reliable results" implies error, but noise in the form of unequal character change probabilities across characters could also result in low support or polytomies, "reliably" conveying the existence of a tough phylogenetic problem. In theory it could also end up increasing hidden support (& congruence w/ a well-corroborated tree) when combined with other partitions (cf. Gatesy & Baker 2005 Syst Bio; Gatesy et al. 1999 Cladistics; Thompson et al. 2012 J Zoo Sys Evol Res).

line 101: "...most transitions to 0" you mean "non-adjacent" state changes, right? Also it might be best to reserve the term "transition" to DNA changes within purines or pyrimidines. Just write "state change" or similar for a morph character.

line 124: Note many divergence estimates from Meredith et al. 2011 are too old and have been superceded by several other, larger datasets (e.g., Dos Reis et al. 2012, Tarver et al. 2016, Phillips & Fruciano 2018). Also more recent studies (Tarver et al. 2016, Esselstyn et al. 2017) are similarly based on more data and show some conflict w/ Meredith (tree shrews, bats, rodent root...), depending on which topology from the latter paper you're using which you've not yet made clear in this MS (e.g., Meredith et al. 2011: fig. 1 based on AAs, or the various AA vs. nucleotide alignments figured in their suppdata file). Crucially, this again underscores the importance of using a well-corroborated tree (see above), not simply one that is "molecular" (a term which spans several decades of fine to bad to awful phylogenetic estimates of various groups).

line 132: use a colon only after a complete clause, so "...clades chosen were AS FOLLOWS:"
Artiodactyla is still valid with the inclusion of whales as sister taxon to hippos, not "Cetartiodactyla" (and line 229; see Asher & Helgen 2010 BMC Evol Biol).

line 315: Jervall not "Jervell"

Figure captions:
The Y-axis "rate" in each graph ("D") is unclear to me. It's labeled as a log scale but spans a heat-map (at right) showing a non-log scale of -2 to -7. Please explain in simple terms what this means and state this in the caption (if possible) or at least refer to the main text (e.g., "... as explained in Methods"). In general the captions are not self-contained, contain acronyms that most readers won't remember ("...the ER model") and thus require flipping back-n-forth to the main text to derive meaning. Please improve.

Note Pattinson et al. is 2015, not 2005, in Fig. 3 caption.

Reviewer 2 ·

Basic reporting

The manuscript was well written with a well laid out narrative that was clear and comprehensible. There were a few typos that I caught and have highlighted in the attached PDF. There was a place in the introduction where I felt the narrative dropped a bit (have noted in PDF) and would suggest a sentence or two to make it feel more cohesive as an introduction.
A suitable array of literature was cited, however, I found a number of issues with the references: some names were misspelled, some citations were missing from the reference list and there were numerous formatting issues/inconsistencies in the reference list. I would strongly encourage the author(s) to invest some time in setting up a reference manager – it’s a slog to set up but very much worth it in the long run.
The article is well structured, with clear aims and nice intuitive figures. The tables are a little harder to comprehend but that is the nature of the results – I appreciated the authors writing out how the values should be interpreted in some instances and would encourage them to so for all the results. It would also be good to have a few sentences in the final paragraph/summary, explicitly linking your results back to the original aims of the paper.

Experimental design

The research article submitted by Brocklehurst and Benevento falls within the aims and scope of PeerJ’s remit. It builds upon the previous study of Sansom et al. (2017) and presents novel and meaningful findings that other researchers will no doubt benefit from. The objective of the study is clearly laid out and the methods are described adequately. That said, I do believe that the methods text would benefit by being fleshed out with a more explicit rationale as to why each metric/analysis has been used, the merits and caveats of each and how the results should be interpreted. By doing so I strongly believe that this paper will be more accessible to those who are perhaps daunted by more method heavy techniques for evaluating phylogenetic datasets.
I also have some concerns regarding the choice of datasets used – namely the morphological datasets for all mammals and artiodactyls (more detailed comments in PDF). There are several published datasets that I would think are better and more suitable, however, I recognise that rerunning and rewriting the current paper may not be feasible (that decision I leave to the authors discretion). If they chose to stick with the current datasets I would suggest expanding the methods/discussion sections to justify their choice more explicitly and discuss the caveats and impacts that it might have on their findings.

Validity of the findings

I have provided more detailed comments pertaining to this section in the manuscript PDF. I have no major concerns but do think that the caveats of your study should be more clearly stated. I have also added a few thoughts and comments in the discussion that can be taken or left as you deem appropriate.

Additional comments

To Neil Brocklehurst and Gemma Benevento,
I enjoyed reviewing your manuscript titled ‘Dental characters used in phylogenetic analyses of mammals show higher rates of evolution but not reduced independence’. It is an area of research I am very interested in and your study builds nicely on the study by Sansom et al. (2017) to address the issues of partitions and biases in phylogenetic data in more detail. It is heartening to know that dental characters are not necessarily as terrible as they seem, or perhaps everything else is just as bad as teeth.... Regardless, I am looking forward to seeing it published in PeerJ soon and being made use of by the research community to improve future phylogenetic analyses.

Annotated reviews are not available for download in order to protect the identity of reviewers who chose to remain anonymous.

·

Basic reporting

The paper is generally well-written throughout, although there are a few minor typos – I have made a few suggested corrections on the attached pdf.

Experimental design

Although the matrices have been taken from published sources, in the interests of reproducibility, I would like the matrices used for the analyses (i.e. after removing selected taxa and characters) and the trees used as “benchmarks” to be included as supplementary data (or uploaded to a repository such as Dryad). Ideally, I would also like the R scripts to be included too.

I have issues with some of the analytical decisions. In particular, I am confused why you have chosen to treat the pectoral and pelvic girdles separately from the forelimb and hindlimb respectively. As you know, the pectoral girdle provides attachment sites for major muscles of the forelimb, and there is a clear functional linkage between the two; for example, the pectoral girdle of arboreal species normally allows considerable freedom of movement at the glenoid, and rotation of the scapula, and is typically combined with adaptations of the forelimb for grasping and pronation/supination movements (see e.g. Argot 2001 – J Morph). Similar functional linkages are seen between the pelvic girdle and the hindlimb (see e.g. Argot 2002 – J Morph). It makes more sense to me to treat the forelimb as including the pectoral girdle, and the hindlimb as including the pelvic girdle. This would also give you a larger number of characters to work with when comparing between partitions. At the very least, I would like to see better justification for why you picked these partitions.

Validity of the findings

The authors engage in some speculation, backed up with relevant citations, on why they observed the patterns they did. However, I think a lot of what they found is the result of particular idiosyncrasies of the matrices they used, and so are not easily generalisable to mammals as a whole. For example, they note that the dental partition for their marsupial dataset shows a lower rate of evolution and greater non-independence (or lower independence) than the dental partitions for the other datasets. However, the dental characters in the marsupial dataset represent a smaller proportion of the total characters than the other matrices (only ~11% compared to 23% in the cetartiodactyl dataset, and 49% in the primate dataset). Furthermore, the marsupial dental characters are relatively “coarse”/high level – 9 of the 29 marsupial dental characters refer to presence or absence of particular loci, which would seem, a priori, to be evolutionarily less labile than the very detailed, “low-level” characters that are much more common in the cetartiodactyl or primate matrices (many of which refer to relatively subtle differences in cusp and crest morphology).

Similarly, the high level of non-independence of dental characters found in the marsupial matrix is likely due to the fact that there are two clades with enlarged gliriform lower incisors (caenolestids/paucituberculatans and diprotodontians), and presence of the enlarged gliriform incisors is associated with a number of additional dental changes, specifically reduction/loss of the other antemolar teeth. This is a much more likely explanation than the proposal that marsupials are somehow “constrained” dentally due to the reproductive mode. The reference to the Werdelin (1987) paper (which is missing from your reference list, incidentally) is a bit misleading, as this was a comparison between Dasyuromorphia and Carnivora only, rather than a comparison across all placentals and all marsupials. In fact, in terms of dental morphology, the order Diprotodontia is probably more variable than most if not all placental orders – it includes forms with bunodont, lophodont and selenodont molars, extreme variation in the morphology of the premolars, and one taxon (Tarsipes) has an almost vestigial postcanine dentition.

There are issues with partitions in some of the other matrices. For example, the very high phylogenetic signal in the soft tissue partition of the primates data matrix is not a “fair”/”honest” attempt to score soft tissues across primates, but is in fact entirely weighted towards characters that support strepsirhine monophyly. Similarly, the soft tissue characters of Bi et al. were clearly chosen to simply reflect the differences in reproductive mode between placentals, marsupials and monotremes, rather than to “honestly” score soft tissue variability across mammals.

I think it would be worth investigating whether there are correlations between your results (e.g. for phylogenetic signal and rate of evolution) and the absolute number of characters in each partition, which would represent a crude measure of sampling “intensity” – I suspect that the high rates of evolution in most dental partitions (with the exception of marsupials discussed above) are because they have been sampled more “intensely” than the other partitions, and so are likely to include very minor, likely highly labile traits. In Cetartiodactyla, the difference between the forelimb (35 characters, ~6% of total) and hindlimb (61 characters, ~10% of the total) may again simply be the result of sampling intensity of the different anatomical regions: nearly 2/3 of the hindlimb characters, for example, are scored off only two bones (astragalus and calcaneus). Testing for correlations between your results and the percentage of the total number of characters each partition represents would also be worth exploring.

Another thing to think about: you carried out tests for character non-independence between pairs of characters. Would it be possible to use the results of these to filter out those characters with the greatest degree of non-independence and then repeat your analyses? How would this affect your results? It would seem, for example, more justified to compare differences in rates of evolution between partitions after removing the characters that show the greatest degree of non-independence.

In summary, I think this paper is worth publishing because it takes a novel, quantitative approach to an interesting and topical question. However, I do genuinely think that many of the differences you have observed are due to idiosyncrasies in the different morphological matrices that make it difficult to compare them “fairly”. I’m therefore not sure whether this is telling us anything about the way that morphological data performs in general. I realise that this is a bit negative, and so I would recommend looking at applying your methods to “phenomic” datasets that have made a fair attempt to score variability across different systems in mammals. One obvious candidate is the O’Leary et al. matrix, although this will undoubtedly have huge amounts of character non-independence due to the decision to score each dental locus separately. Another one that you might consider is the primate matrix of Ni et al. (2013 – Nature), or subsequent updates of this matrix – it represents a fair attempt to score variability across the entire primate skeleton and also soft tissues, and so might be worth exploring for future studies.

In the interests of openness, I would like to sign my review, and I am happy for the authors to contact me if they require clarification of any of the points I have raised.

ROBIN M. D. BECK
Lecturer in Biology
School of Environment and Life Sciences
University of Salford, Manchester M5 4WT
T: +44(0) 0161 295 4994
Email: r.m.d.beck@salford.ac.uk

---

## Round 0.2 · Minor Revisions

Dear authors,

I have accepted the decision of ‘minor revisions’ from the reviewers.

I look forward to receiving your revised manuscript.

Reviewer 2 ·

Basic reporting

The authors have done a good job addressing the comments brought forward in the previous round of reviews. In this version of the manuscript, I have no major concerns although note there were a few typos that I caught and have highlighted in the attached PDF. I also found numerous inconsistencies in the reference list that require attention.
It would also be useful to have a list of the supplementary materials provided somewhere to make navigation of files simpler rather than just file names especially as some of the supplementary files do not include titles

Experimental design

I appreciated the authors taking on board my previous comments regarding expanding out their methodology and substituting out/adding to the datasets used in the analyses. I have no further comments.

Validity of the findings

I am still wary of the use of the Bi et al. matrix in the study. I recognise the authors reasoning for using this dataset and that they have included a biref statement in the methods section regarding the limitations of this dataset, but, I do feel there could be more information provided in the discussion (noted in annotated PDF attached with my review) following on from the comments provided by myself and the other reviewers in the previous round of reviews. I appreciate that this is a meta-analysis but at the same time, I do think it is important to be very clear about the limitations of the input data and what effect they could (and likely are) having on the results and conclusions of this study. In the present draft of this manuscript, these issues are mentioned but only briefly and, in my opinion, not given the weight they deserve.

Additional comments

Below are a few comments on the 'Response to reviewers' document. (Apologies, I am working in transit so is easier to copy and paste directly into this section)

R2 “I am not sure why you're comparing it [the Meredith tree] to Bi et al. which is targeting euharamyid anatomy (i.e. mostly fossils) and derived primarily from Zhou et al. 2013. Surely, something like O'Leary et al. The placental mammal ancestor and the post-K-Pg radiation of placentals (or derivatives thereof) which includes a greater number of extant taxa, would be more appropriate?

NB We rejected the O’Leary dataset for two reasons. One is that identified by reviewer 3; the matrix uses some extremely unusual practices in formulating characters that are neither reprehensive of usual procedures in phylogenetics nor particularly informative about evolutionary processes. The second is that a lot of the characters, particularly dental characters relating to number of teeth in particular regions, are scored as unknown for large numbers of taxa. Its isn’t clear quite why this is the case, but it means that the full array of morphologies cannot be analysed.

R2 Note that the missing scores in the Tree of Life datatset are the result of where decision cannot be made as to which tooth/teeth are present or lost in a series. The authors scored by tooth position – there are reasons both for and against this and for scoring just by number of teeth in a series. Both matrices have unusual practices for formulating characters. But ultimately, the O’Leary matrix produces a seemingly more accurate phylogeny than the Bi matrix for a ‘better’ sample of mammals. I appreciate your reasoning for using the Bi matrix and do not think it requires changing for this publication although moving forward it could be interesting to compare different Mammalia matrices using the methodology outlined in this study to see if there are consistent signals in character independence etc.
* * *
R2 “The inclusion of fossils does tend to improve the accuracy of phylogenies, and the lack of fossils in the O'Leary dataset is an issue but perhaps not for your purposes. Could also look at Halliday et al. (Resolving the relationships of Paleocene placental mammals) although that dataset has a whole host of problems so may be better going back to sources datasets of that study (e.g. Wible et al. 2009).”

NB As we are analysing the evolution of characters over a molecular tree, fossil taxa are dropped from the matrix. Therefore the substantial inclusion of fossils in the O’Leary et al. and Halliday et al trees is not helpful for our analyses.

R2 I referring to the inclusion of fossils in the construction of matrices (apologies that that was not clear). I know the presence/absence of fossils is irrelevant for your analyses but if they improve the quality and accuracy of the dataset, both in how it is constructed and the results it produces, then using such a dataset would provide more credence to your meta-analysis and conclusions.

Annotated reviews are not available for download in order to protect the identity of reviewers who chose to remain anonymous.

·

Basic reporting

A few additional typos need correcting - see attached marked up Word document.

Experimental design

This has been improved from the previous iteration, and I don't have any substantive comments.

Validity of the findings

no comment

Additional comments

The authors have done a good job at responding to my comments/suggestions and those of other reviewers. There are a few minor issues that I would like to be addressed in the revision, but these should be relatively easy to deal with. I have picked up some further typos in the revised text (including in the reference list) - see the MS document submitted with my review.

My major outstanding issues that I would like the authors to address are as follows:

1. I didn't pick this up before, but the authors emphasise that they used time-scaled/time-calibrated molecular phylogenies to compare their morphological characters against. However, what is the relevance/importance of the fact that the molecular phylogenies have divergence dates associated with them? Was this information used in the calculation of rates of evolution?

2. As I mentioned in a previous review, the soft tissue partition for the Bi et al. (2014) matrix is definitely not a "fair" reflection of soft tissue variability among living mammals: there are only 3 characters, and all of them relate to details of the reproductive system and have pretty obviously been selected to distinguish placentals from marsupials and monotremes. This should be acknowledged somewhere, and might explain why the violin plots for this partition are so odd looking.

3. A related point: for the violin plots, I recommend including in the labels the number of characters in each partition, e.g. "dentition (n = 100)" or "dentition (100 characters)" or something like that. This would enable readers to see whether the more "extreme" violin plots are for partitions with only a few characters in them.

4. The authors make a very interesting point about the possibility that the increase in rates of dental evolution in the artiodactyl dataset might be due to the large number of herbivorous taxa. This is a nice hypothesis that would definitely be worth testing further in future, but I would like the authors to expand this a bit and explain in more detail why this might be the case. How exactly have plants changed more than “meat” in terms of functional requirements? I am not an expert in this, but I would assume that the functional requirements for eating a leaf would be the same regardless if it was a Palaeocene or Pliocene leaf. However… clearly the evolution of abrasive grasses does represent a major change that likely led to a major change in dental structure of mammals, with the evolution of increasingly hypsodont and in some cases hypselodont teeth. Having said that, some authors have argued that exogenous grit has been the major driver of hypsodonty in mammals (see e.g. Madden’s book “Hypsodonty in Mammals”).

---

## Round 0.3 · accepted · Accept

Dear authors,

I have accepted the reviewers decision to 'accept' your manuscript for publication in PeerJ. One reviewer noticed two small issues, but they can be corrected in proof.

You will soon be contacted shortly by the production staff to take you through the proof stages.

Congratulations again, and I hope you will consider PeerJ as your publication venue in the future.

Reviewer 2 ·

Basic reporting

See General comments to the author section. Previous comments in this section have been addressed and in my opinion, now meet the requirements for publication.

Experimental design

See General comments to the author section. Previous comments in this section have been addressed and in my opinion, now meet the requirements for publication.

Validity of the findings

See General comments to the author section. Previous comments in this section have been addressed and in my opinion, now meet the requirements for publication.

Additional comments

I am happy with the changes the authors have made to the current manuscript and find that they have addressed my comments more than adequately.

Two tiny things:
In the manuscript references, there is an erroneous comma after Marjanovic in Marjanovic & Laurin 2019
In the supplementary tables (word doc) there is the use of cetartiodactyls in the caption for Table S1.

Otherwise, I endorse the acceptance and publication of this manuscript in PeerJ.

·

Basic reporting

Unchanged from previous version

Experimental design

Unchanged from previous version

Validity of the findings

Unchanged from previous version

Additional comments

The authors have responded to my few minor points appropriately - I'm now happy to see this published. Congratulations!